# Assessment of Anxiety and Depression in Polish Primary Parental Caregivers of Children with Cerebral Palsy Compared to a Control Group, as well as Identification of Selected Predictors

**DOI:** 10.3390/ijerph16214173

**Published:** 2019-10-29

**Authors:** Barbara Gugała, Beata Penar-Zadarko, Danuta Pięciak-Kotlarz, Katarzyna Wardak, Aneta Lewicka-Chomont, Magdalena Futyma-Ziaja, Józef Opara

**Affiliations:** 1Medical College of Rzeszów University, 35-959 Rzeszow, Poland; beata.penar@yahoo.com (B.P.-Z.); werhub0@tlen.pl (D.P.-K.); wardakk@interia.pl (K.W.); aneta.lewchom@gmail.com (A.L.-C.); magdafutyma@gmail.com (M.F.-Z.); 2Center for Innovative Research in Medical and Natural Sciences, University of Rzeszów, 35-959 Rzeszow, Poland; 3Academy of Physical Education in Katowice, 40-065 Katowice, Poland; jozefopara@wp.pl

**Keywords:** anxiety, depression, determinants, parents of children with cerebral palsy

## Abstract

Background: Taking care of a child with Cerebral Palsy (CP) may be linked with adverse effects in the parents’ physical and mental health. The causes of anxiety and depression symptoms associated with childcare are still not fully understood. Aim: To assess the intensity of anxiety and depression symptoms in parents of children with CP compared to a control group and to identify selected mental health predictors. Design and Methods: Data were collected from 301 respondents, including 190 parents of children with CP (study group) and 111 parents taking care of children developing normally (control group). Intensity of anxiety and depression was rated using the Hospital Anxiety and Depression Scale (HADS) scale. Gross Motor Function Classification System for Cerebral Palsy (GMFCS), Sense of Coherence Scale (SOC-29), Berlin Support Social Scales (BSSS) scales and a specially designed questionnaire were used to assess the predictors. The investigated variables included the children’s and the parents’ characteristics, as well as environmental factors. The analyses applied Spearman’s rank correlation coefficient, M(SD) as well as multiple regression. Results: The level of anxiety and depression was clearly higher in the parents of children with CP–the mean levels of anxiety and depression in the study group and the controls amounted to 8.1 vs. 4.7 and 6.8 vs. 3.7, respectively. The factors associated with intensity of anxiety and depression in the parents of children with CP included lack of social support, mainly perceived and received support, unsatisfying parental health status, poor economic status of the family, as well as difficult living conditions, sense of coherence, loneliness, the parent’s gender, and the child’s intellectual disability. Conclusions: Identification of significant anxiety and depression predictors, understood as modifiable factors, should be considered in determining and planning comprehensive support for a child with CP and his/her primary parental caregiver.

## 1. Introduction

In most families a birth of a child is associated with immense joy and happiness, but also with a need to adjust to new obligations and tasks, in addition to those which existed earlier. Over time, the situation becomes normalized in families with healthy children, however if physical or intellectual disabilities are identified in the new family member, the parental duties take on a completely different meaning. Many parents cope well with their new responsibilities, from the start after the child’s disability has been diagnosed, however it has also been reported in the literature that the obligation to take care of a child with a disability may become arduous, adversely affecting the caregiver’s physical and mental health, and ultimately their sense of well-being [1]. The most frequent developmental disorders observed in children include Cerebral Palsy (CP) defined as non-progressive disorder of motor functions, resulting from impairment of upper motor neuron initiated during embryonic or fetal development or during the perinatal period [2]. The prevalence of the condition ranges from 1/1000 live births for children born at term to 2.1/1000 live births for children born preterm [3]. The epidemiologic trends are similar in the developed and developing countries [4], with a tendency towards higher rates in families with low economic status [5]. Due to the high prevalence, the syndrome is the most common cause of physical disability and, preceded only by intellectual disability, the second most frequent cause of persistent neurodevelopmental impairments in children [6,7,8,9]. In clinical and epidemiological terms, CP is not a uniform condition. The main factor determining the clinical picture at a later stage is the degree of development and maturity of the central nervous system at the time the adverse factors operate. Motor disability is associated with paresis of the limbs, involuntary movements, as well as impaired balance and motor coordination. The symptoms first appear in early childhood and persist during the patient’s entire life. In addition to these motor defects, there are many sensory and communication-related problems, as well as secondary musculoskeletal impairments [10,11]. The extent of support required to satisfy the child’s biological and psychosocial needs depends on the severity of the disability, as well as the limitations faced by the child in activities of daily living. In his/her daily life a person responsible for a child with CP must cope with the child’s motor and sensory disabilities and, in addition to that, he/she must enable implementation of a wide range of necessary medical interventions and rehabilitation. The main challenges faced by the parents include effective management of the child’s health problems and a need to satisfy the defined standards of care [12]. Long-term responsibility for a disabled child, combined with a necessity to face adversities and challenges associated with handling the child, frequently leads to negative emotions, such as anxiety, sadness, anger, hopelessness and distress, experienced by the person, and adversely affecting the functioning of the parents and the entire family [13,14,15]. The most common effects of long-lasting caregiver burden include depressive and anxiety disorders of varied intensity [1,6,16,17,18,19,20,21,22,23,24,25,26]. However, intensity of anxiety and depressive symptoms in the parents is not related exclusively to predictors associated with the child’s disability, but is also linked to personal, social and economic variables resulting from the parents’ needs and health status as well as factors occurring in the family; this fact reflects the multidimensional effects of these determinants in health-related quality of life (HRQOL) in individuals taking care of children with CP [27]. The subjective perception of caregiver burden depends on many factors, such as the individuals’ life situation, personality and the way they perceive the experience and what meaning they attribute to it. As suggested by theories of stress and coping, the effects produced by stressors linked with handling a disabled child in the parents’ health depend more on their personal resources than on objective requirements faced by them, or social resources available to them. In the salutogenetic model, the sense of coherence is treated as a superior individual resource, essential for maintaining psychophysical health, while social support is seen as a potentially available protection received from the individual’s social networks. These personal resources are used in the process of coping with difficult and challenging situations and are understood as preventive factors making it possible to maintain health [28].

The study was designed to assess the intensity of anxiety and depression symptoms in primary parents of children with CP, compared to a control group, and to comprehensively investigate the factors associated with the intensity of mental health problems in the study group. Comprehensive investigation of the factors involves assessment of predictors linked to the child, the parent and the environment.

## 2. Materials and Methods

### 2.1. Settings and Participants

The study involved 301 respondents, including 190 primary parental caregivers of children with CP (the study group) and 111 primary parents of normally developing children (control group), and was carried out from May 2013 to June 2016. The subjects were recruited for the study group among families of children receiving specialist services in regional rehabilitation facilities. The controls were recruited among parents visiting primary care physicians with their children due to transient health problem such as a cold, or to receive preventive vaccination. The researcher personally contacted all the eligible parents, in the facilities, in order to conduct the survey, and to obtain informed consent in writing from those willing to participate. The respondents completed the questionnaires on their own, in the facilities which they visited with their children.

### 2.2. Eligibility to Participate in the Study

Recruitment into the group of respondents was based on established inclusion criteria. Common to both groups were: declaration that the person is the primary parent of the child (based on the hours of care provided), permanent residence with the child, no financial compensation for the care, no mental illness diagnosed and no antidepressant treatment, as well as a written consent to participate in the study. Additional criteria for inclusion into the group of parents of CP children were as follows: CP diagnosed in accordance with ICD-10, care provided to children aged from 2 to 18 years. The inclusion criteria for the parents of children developing normally were as follows: lack of a diagnosed chronic disease or physical and mental disability in the child, care provided to children aged from 2 to 18 years. A specially designed questionnaire was used in both groups to identify the socio-demographic characteristics of the families, the primary parents and their children. Out of those 242 subjects participating in the recruitment procedure, 40 individuals were not qualified for the study group, as they failed to meet the inclusion criteria defined for parents of children with CP (25 individuals were not the primary parents, in three cases—antidepressant treatment was used, 13 individuals did not agree to participate in the study). After the questionnaires were returned, 12 were rejected due to missing sociometric data and other incomplete responses. During the recruitment for the control group, 131 individuals were qualified to participate, however 10 individuals refused to take part due to a lack of time, lack of willingness to fill in the questionnaire and for other reasons; five individuals were not qualified because their children were diagnosed with chronic diseases. It was assumed that correctly completed surveys were those in which responses were given to all the questions. Ultimately, the study group consisted of 190 parents and the control group comprised 111 subjects, who met the inclusion criteria and provided informed consent to participate in the study. All the subjects and controls taken into account in the study were biological parents of the children. The response indexes in the study group and in the control group amounted to 78.5% and 84.7%, respectively.

A set of questionnaires was handed to the subjects in paper form, and time needed for completing the survey ranged from 15 to 30 min during a visit to the hospital. To ensure the participants’ anonymity, the completed surveys were kept separately from the consent documents. The parents who expressed willingness to participate were informed about the main purpose of the study, about importance of their contribution, confidentiality of their input, duration of time needed for completing the survey and issues addressed by the questionnaire. The subjects qualified for the study were not paid for completing the questionnaire.

### 2.3. Ethical Considerations

The study was approved by the Bioethical Commission at the University of Rzeszów (KB/3/06/2013) and was conducted in compliance with the Helsinki Declaration [29].

### 2.4. Clinical Parameters

Data related to the study group were retrieved from the patient records kept at the hospital ward, and included diagnoses of the type of CP, assessment of motor capacity with Gross Motor Function Classification System for Cerebral Palsy (GMFCS), and evaluation of intellectual impairment as defined by the Diagnostic and Statistical Manual of Mental Disorders (DSM-5).

GMFCS—the scale assesses independent movement, most importantly focusing on the sitting position (trunk control) and gait. The evaluation system consists of 5 levels (from I–V) in five age groups, i.e., <2 years; 2–4 years; 4–6 years; 6–12 years; and 12–18 years of age. Distinctions between levels of motor capacity are based on limitations in independent mobility, as well as a need to use support devices and wheelchairs [30].

DSM-5—according to the Manual, intellectual disability is characterised by both intellectual deficit as well as deficit in adaptive functioning which starts during the individual’s development. Diagnostic classifications distinguish four levels of intellectual disability, i.e., mild, moderate, severe and profound [31].

### 2.5. Measures

In accordance with the main purpose of the study, assessment focused on the intensity of anxiety and depression in the parents of children with CP, compared to the parents of normally developing children. Intensity of anxiety and depression in the subjects was assessed using the Hospital Anxiety and Depression Scale (HADS).

#### 2.5.1. HADS

A questionnaire developed as a tool to identify anxiety and depressive disorders in various health conditions and in the general population. It contains 14 questions included in two “hidden” subscales: anxiety (seven questions) and depression (seven questions). Answers are given on a four-point scale from 0 to 3 points, with the scores in each subscale theoretically ranging from 0 to 21 points. A score above 10 points indicates an anxiety disorder or depression. For the Polish version of the scale, the value of Cronbach’s α coefficient for the whole anxiety and depression subscale is 0.81–0.89 [32,33].

The tools enabling assessment anxiety and depression determinants, i.e., the factors linked to the child (GMFCS, DSM-V), to the main caregiver (Sense of Coherence Scale (SOC-29)) and the environment (Berlin Support Social Scales (BSSS)) were all available in the Polish language. The research tools also included:

#### 2.5.2. Specially Designed Questionnaire

Both groups were assessed for sociodemographic characteristics of the primary parents, their families and children. The rating of the economic status was based on the previous 12 months which the respondent was asked to assess by selecting one of the three responses describing their economic security as: good, if the family had enough money for essentials, for the child’ rehabilitation and to save; mediocre, if the available resources were sufficient for essentials but not for saving; and poor, if the family’s resources were insufficient for basic needs.

The part of the survey focusing on loneliness defined the related states and asked the respondents to specify how frequently each state was experienced by them during the previous 12 months; there were three possible answers: often, occasionally, never. The specific question was: how often in the previous 12 months did you experience: (1) a feeling of not being understood; (2) a sense of being rejected by others; (3) a sense of discrepancy between expectations and actual relations; (4) a sense of lack of connection to the loved ones; (5) a feeling that you cannot establish emotional closeness; and (6) a feeling you do not belong to the society. The obtained responses were scored by awarding two points when the symptoms occurred often, 1 point—occasionally, 0 points—never [34]. Satisfaction with one’s health status was rated on a five-point Likert scale, where the score of 5 corresponded to “very satisfied” and 1 to “very unsatisfied”. The obtained scores were used in statistical analyses, as dependent variables, in identifying determinants for intensity of anxiety and depression. The remaining tools applied in the study included such standardized tools as:

#### 2.5.3. Sense of Coherence Scale (SOC-29)

Tool assessing personal resources and a sense of coherence, contains 29 questions, consisting of three subscales: I—comprehensibility (11 questions); II—manageability (10 questions); III—meaningfulness (eight questions). Answers are given on a 7-point scale with the described extremes. Average scores in the subscales gave the overall result for the sense of coherence. The theoretical spread of the results ranges from 29 to 203, classifying the results at low, medium and high levels. The scale is characterized by good psychometric properties- high accuracy and reliability. For the Polish version of the scale, the value of the Cronbach’s α coefficient is 0.87 [35].

#### 2.5.4. Berlin Support Social Scales (BSSS)

The social support measurement tool contains 6 independent scales with 32 questions contained in 4 scales: perceived support (eight questions), need for support (four questions), support seeking (five questions), actually provided and received support (15 questions). Answers are given on a four-point Likert scale; where 1 means- completely disagree, 2—somewhat disagree, 3—somewhat agree, 4—strongly agree. A higher score, in the range from 1 to 4, means greater social support. The coefficient of internal cohesion of Cronbach’s α for subsequent scales is: 0.83, 0.63, 0.81, 0.83 [36].

### 2.6. Statistical Analyses

Arithmetic means, medians, interquartile range and percentage fractions were used in the statistical description. The analysis was carried out taking into account the numerical values of HADS measures. Statistical significance of the differences between the groups was evaluated using the Mann–Whitney test. Analysis of the sociodemographic data with respect to the study group and the controls involved comparing the information on the percentage distribution of the relevant data in the two groups. Significance was assessed with Chi-square test of independence. The scores in the HADS scale were examined for the effects of such factors as the sense of coherence, age and sense of loneliness, social support and the children’s intellectual impairment. For this purpose, Spearman’s rank correlation analysis was used, with a significance test applied to the correlation coefficient (*p*). Multiple regression analysis was also applied, where the optimum regression model was determined using the procedure of forward stepwise regression. Statistical significance was assumed for the values of *p* < 0.05. The analyses were performed using Statistica v. 10 software.

## 3. Results

### 3.1. Socio-Demographic Characteristics of Primary Parents

The socio-demographic data of both groups of parents and their children as well as the characteristics of the families, with a comparison between the study group and the control group, are presented in Table 1 and Table 2. As regards the sociodemographic data, there were no differences between the study group and the control group in terms of age, place of residence, type of family, religiosity, number of children in the family and the number of individuals taking care of the child. Six subjects described themselves as non-believers, the others reported they were Catholics. There were, however, differences in formal education, occupation, satisfaction with one’s health status, economic security, housing conditions and the sex of the parent participating in the survey.

### 3.2. Descriptive Analysis of Study Variable

Analysis of the results showed that the level of anxiety and depression is definitely higher in the group of parents of children with CP. Table 2 presents distributions of these measures in the two groups, by showing the basic values of descriptive statistics (mean, median, standard deviation and 95% confidence interval). The parents of the children with CP present significantly higher levels of anxiety and depression.

### 3.3. Presentation of Measures Linked to the Condition of the Child in the Study Group

The children of the parents in the study group were all diagnosed with CP. In terms of sex, majority of the children were male, and the most common coexisting symptoms included speech and balance disorders and intellectual disability. In the motor assessment (GMFCS) most children were classified at level 2 (Table 3).

### 3.4. Linear Regression Analysis of Study Variable

Comparative analysis of the characteristics of the two groups, as presented in Table 1, shows rather clear differences between the groups related to the parents’ education, financial status and other factors. Therefore, it is possible that the intensity of anxiety and depression in the study group may be linked with the different living conditions rather than only with the management of the child with CP. Regression analysis was carried out in order to assess effects of handling a child with CP in the intensity of anxiety and depression, controlled for demographic factors. The constructed regression models applied the HADS-A and HADS-D measures as a dependent variable, while the initial set of independent variables comprised all the demographic factors listed in Table 1, and the variables determining allocation to the study group or the control group. To facilitate interpretation, most of the demographic factors considered were shown in a dichotomous form.

The procedure of forward stepwise regression was applied to select the factors significantly affecting the measures of anxiety and depression. The results of regression analysis are shown only for those factors which produced statistically significant effects.

The regression analysis confirmed there was a difference in the intensity of anxiety and depression between the parents of the healthy children and the parents of the children with CP and showed the difference was similar (approximately 2.8 points in HADS-A and 2.7 points in HADS-D), also when demographic factors were controlled, and the results from Table 2 were verified. Furthermore, the regression analysis shows that the level of anxiety increases with poorer health status of the parents of the children with CP, as well as poorer financial status of the family. The level of depression, on average, increases by 1.85 points, with poorer health status reported by the parent, and by 2.03 points when the child with CP does not have his/her own room (Table 4 and Table 5).

The purpose of the further analyses was to identify factors producing statistically significant effects in the level of anxiety and depression in the parents of children with CP (therefore the following analyses are related only to the study group). The factors taken into account in particular included: intensity of symptoms (GMFCS—with dichotomous distinction of two levels), level of social support, as well as demographic factors considered in the earlier analyses. The procedure of forward stepwise regression was applied to build a regression model (as presented in Table 4 and Table 5). It was found (Table 6) that the model of regressions describing the level of anxiety in the parents of the children with CP comprised: the parent’s gender, financial security and perception of available support.

The level of anxiety is not affected by the child’s motor disability assessed with GMFCS, or by the other measures of support, and other demographic factors (Table 1). One-point increase in social support rating on average corresponds to 1.39 points decrease in the level of anxiety. The level of anxiety on average is 1.11 point higher in mothers, compared to fathers, and poor financial status coincides with higher level of anxiety, on average by 1.24 points.

The level of depression is related to two measures of social support, as well as the self-reported financial status of the family and self-assessed health status of the primary parent. The model for the measure of HADS-D explains greater rate of variability in this measure (*R*^2^ = 30.1%), compared to the model or HADS-A (*R*^2^ = 11.4%) (Table 6 and Table 7).

### 3.5. Analysis of The Relationship Between Factors Linked to Parents of Children with CP and Intensity of Anxiety and Depression in the Study Group

Further analysis of factors significantly related to the intensity of anxiety and depression in the parents of the children with CP was carried out using the Spearman rank correlation coefficient. Negative results of the correlation coefficients suggest that the level of anxiety and depression decreases with a higher sense of coherence, and lower level of anxiety corresponds to older age of the subjects. A positive value of the correlation coefficient reflects a tendency for depression to increase with the parents’ age and for anxiety and depression to increase with the subjects’ growing sense of loneliness. A statistically significant correlation is also observed between the child’s intellectual disability and the parent’s anxiety. The greater the child’s intellectual impairment, the more likely it is for the parent to present a higher level of anxiety. The environmental factors subjected to the analyses include social support, which is significantly linked to the scores on the HADS scale. The analysis of Spearman’s correlation showed the weakest associations between the sense of anxiety and support needed, support perceived and support seeking. The value of the correlation coefficient is very important; the level of anxiety (and depression in particular) corresponds to poorer rating of support, particularly in terms of its availability. Assessment of depression showed that in the group of practicing Catholics more than one in two subjects had normal scores, compared to only one in three subjects in the other group. Analysis of the respondents’ attitude towards religion, where they were divided into practicing Catholics and non-practicing Catholics, showed that depression was more common in the latter group. Assessment of physical capacities, based on the GMFCS scale, showed that this factor was linked to the level of depression in a way approaching statistical significance. A lower score on GMFCS corresponded to a slightly higher level of depression (Table 8).

## 4. Discussion

Research on parents of children with CP focuses mainly on the impact of care on the health of parents as well as on assessment HRQOL. A review of the related literature and published findings shows that parents of children with CP tend to present a higher level of distress and poorer mental health [6,21,37,38,39,40], and that anxiety and depression are the most common mental health disorders [1,6,41,42]. However, the determinants of anxiety and depression are still not fully understood, therefore the related research needs to be continued.

The present study investigates intensity of anxiety and depression in the primary parents of children with CP in comparison to the control group, with a focus to selected determinants of these mental health disorders. The level of anxiety and depression was visibly higher in the group of parents of children with CP, the means identified for anxiety in the study and control groups being 8.1 vs. 4.7 and for depression 6.8 vs. 3.7, respectively. As shown in the results section, the most important predictors of anxiety and depression in the current study included the lack of social support, mainly perceived and received, dissatisfaction with one’s health status, poor economic status of the family, as well as difficult living conditions of the family. The sense of coherence, loneliness, the parent’s gender, as well as the child’s intellectual disability were also associated with the intensity of anxiety and depression. Sajedi et al., in a study focusing on mothers of children with CP, found that the rate of depression was 2.26 times higher in the study group compared to the controls (mothers of healthy children). Assessment of the related causes showed that the prevalence of depression is not related to the type of CP [41]. The current findings suggest that the mothers presented with 1.11 points higher level of anxiety than the fathers. This finding is consistent with the results reported by Byrne et al. who observed that mothers of children with CP had poorer mental and physical health than fathers because they spent more time taking care of their children than the fathers [17].

The above is also supported by the findings published by Derajew et al. who observed that the risk of depression was higher in females taking care of patients [43]. The above suggests that preventive measures related to depression should particularly focus on mothers of children with CP. The relation between motor capacities assessed with GMFCS and the level of depression approaches statistical significance. A lower score on the GMFCS corresponds to a slightly higher level of depression. As reported by other authors, in most studies this variable was not positively correlated to anxiety and depression in parents of children with CP [21,25,26,41,43,44,45].

In the present study, a child’s greater intellectual disability is linked to a higher likelihood of anxiety in the parent. Other authors, such as Parkes et al., point out that intellectual disability is most closely related to parental stress [39]. Quality of Life Inventory-Disability (QI-Disability), being a credible and valid measure of quality of life, applicable to the entire spectrum of intellectual disability, has the potential to enable identification of support needs and assessment of response to interventions administered [45].

It was observed that the number of people taking care of a child with CP is clearly linked to the level of depression because the HADS scores tend to decrease as the number of individuals taking care of a child increases. In a study by Park and Nam, the parents spent about 14–15 h a day looking after a child with CP, without seeking support from other family members. The findings showed no effect of the time dedicated to care in the prevalence of depression, however increasing duration was associated with a greater risk of exceeding the threshold of depression [46]. The above results were confirmed by Sawyer et al. who reported that time pressure significantly influenced the occurrence of depression among parents of children with CP [25]; likewise, Roxburgh established that time pressure predicted the occurrence of depression [47]. Increased caregiver burden, associated with a necessity to deal with the child’s disability and health problems, additionally determines the time demand and emotional involvement of the caregiver [48]. Given the above, in order to relieve the caregiver burden and improve parents’ mental health it is necessary to create a more effective social support system comprising family, relatives, and neighbours offering emotional and practical assistance in solving everyday problems. Psychosocial interventions should comprise information on coping with time constraints and emotional burden. Poor psychological condition of parents may adversely affect their ability to manage the development and rehabilitation of the children, therefore, by improving and protecting the parents’ mental health it will be possible to significantly facilitate the treatment and rehabilitation of their children with CP.

Studies by Cheshire et al., and Unsal–Delialioglu et al. show that depression is an important prognostic factor for caregiver burden, reflected by a significant positive linear relationship [6,26]. Marrón et al. assessed determinants of caregiver burden in a group of 62 mothers and fathers of children with CP and reported that the most important predictors of the burden include the degree of the child’s disability, depression and a sense of competence. Anxiety was not identified as a predictor of the burden [49]. This is consistent with the results reported by Ones et al. who assessed 46 mothers of children with CP and 46 mothers of healthy children and did not find differences in the level of anxiety between the two groups. These authors concluded that parental anxiety may be more related to unexpected life events than to the chronic effects of the factors recognized as predictors. Persisting impact of predictors can lead to depression symptoms which are more noticeable in long-term assessment of human behavior [50].

An increase in the level of coherence corresponded to lower level of anxiety and depression in the subjects. Earlier studies focusing on parents of children with developmental disorders report that these parents predominantly present poorer sense of coherence, compared to parents of children developing normally, which is associated with their tendency to use avoidant strategies in coping with stress, and with greater prevalence of anxiety and depression in these individuals [51,52]. In the study by Ketelaar et al. parents with a high sense of competence presented better physical and mental health, better relationships and greater satisfaction in personal relationships, unlike those with poor sense of competence, which was significantly correlated to anxiety and depression [53]. In terms of education, compared to the control group, the study group included more people with lower education. In the analysis of predictors, it was found that education was linked to depression in the study group. In both groups the parents’ age was comparable; the findings show that a risk of depression increases with the parents’ age. Generally, more common in the latter group of the subjects, a greater risk of anxiety and depression corresponded to more severe feeling of loneliness. A study by Chang–Kyo showed that caregivers’ quality of life was significantly affected by a combination of the general characteristics related to the caregiver such as age, health, religion, duration of care, and relations with the child [54].

Analysis of religion-related effects in the respondents, divided into practicing Catholics and non-practicing Catholics, showed that the latter were more at risk of depression. This could be explained by the fact that in order to overcome negative feelings and thoughts about the child’s illness and future, and wanting to cope with the related challenges, parents often maintain close contact with God and spiritual values, providing good support in the process of accepting the child’s chronic condition [55].

In addition to spiritual support, parents of children with CP also need other types of assistance to function effectively. In the present study, the factor of social support can mainly be linked to the scores on the HADS-D scale. More severe depression corresponded to poorer rating of available and received support as well as decrease in support seeking. This finding is disturbing from the viewpoint of the child’s opportunities for development as well as the parent’s well-being, because a lack of interest in support and a failure to seek help leads to irreversibly lost chances for the child’s development. Taking care of a child with CP is linked with a need to constantly seek support to improve the child’s living standard and to aid his/her natural development, despite the chronic condition. In their study, Davis et al. concluded that care of a child with CP may favourably affect the parent’s ability to build new social support networks, however when no external support is sought, the challenging situation may negatively impact the caregivers’ physical health, social relations, freedom and independence, as well as family relationships and financial stability [1]. Furthermore, important findings reported by Chih and Liang in their 2017 study showed that a risk of distress experienced by family members is increased by unmet medical needs due to high costs [56]. A literature review and meta-analysis of the related findings showed that interventions involving parents significantly improved their HRQOL, particularly if these were delivered in direct contact with the caregiver (face-to-face mode) [57]. Keeping this in mind, family-centred services should offer social support and develop various strategies to deal with the challenges and needs of CP parents [58]. Furthermore, the identified predictors of anxiety and depression, as well as the effects of these conditions, understood as modifiable factors, should be taken into account in designing comprehensive support for a child with CP and his/her primary parents.

Vogts et al. carried out a study in a group of parents of children with CP, using Craig Hospital Inventory of Environmental Factors (CHIEF) questionnaire, and they concluded it is necessary to develop research approaches and tools which will allow to assess barriers to participation in children with CP, taking into account socioeconomic factors and other important information related to their functioning [59]. On the other hand, Sciariti et.al. have proposed electronic implementation of International Classification of Functioning, Disability, and Health (ICF) Core Sets for children and youth with CP, arguing that ICF Core Sets for CP will be highly practicable in management of health services and in building profiles of functioning and disability for patients worldwide [60]. By recognising the needs of a child with CP and evaluating his/her functioning in the society it will be possible to facilitate management of services and to relieve the caregivers.

Assessment of primary parents of children with CP for their mental health should be a key element of routine screening in order to identify risks and implement preventive measures or treatment of mental disorders, to improve the quality of life for the families.

## 5. Conclusions

Compared to the control group, the parents of children with CP presented visibly higher level of anxiety and depression. Based on multidimensional variables, including characteristics of both the children and the parents, as well as environmental factors, it was shown that the most important determinants of this condition include parental sense of coherence, loneliness, gender, self-reported health status, economic status of the family, living conditions as well as social support perceived and received. A forward stepwise regression analysis confirmed the multifactorial relationship between the determinants and intensity of anxiety and depression in the parents of children with CP. Therefore, awareness of the factors contributing to the intensity of anxiety and depression will allow healthcare professionals to adequately plan measures designed to minimize the negative effects of caregiver burden and to improve functioning of parents and families of children with CP.

### 5.1. Limitations

The limitations of the current study include the fact that it focused on subject recruited only in regional rehabilitation facilities in south–eastern Poland. Another limitation is the small size of the study group. In order to obtain more general findings, it would be worthwhile to carry out similar research involving a larger group of subjects from the whole country.

### 5.2. Practice Implications

Our study contributes new evidence to the literature suggesting that higher intensity of anxiety and depression in parents of children with CP is a major public health problem which should be addressed by focusing on issues related to management of healthcare, social work and welfare. The present findings reflect a problem to which medical personnel and social workers should pay more attention. In order to reduce caregiver burden and decrease symptoms of depression in parents of children with CP, it is necessary to consider measures which could effectively improve social support. Furthermore, in addition to institutional social support networks it would be worthwhile to consider building formal support networks involving families and friends of families with children with CP.

## Figures and Tables

**Table 1 ijerph-16-04173-t001:** Sociodemographic and family characteristics.

Variable		Study Group Me (IQR, R) ^1^	Control Group Me (IQR, R) ^1^	*p* ^2^
Age of parents (yrs.)		40 (35–46, 22–69)	39 (34–44, 24–58)	0.158
		N (%)	N (%)	*p* ^3^
Sex of the Parent	Female	138/(72.6)	97/(87.4)	0.002
	Male	52/(27.4)	14/(12.6)	
Place of Residence	Rural Area	115/(60.5)	59/(53.2)	0.211
	Urban Area	75/(39.5)	52/(46.8)	
Education	Vocational	70/(36.8)	20/(18.1)	0.000
	Secondary	74/(38.9)	42/(37.8)	
	Higher	46/(24.2)	49/(44.1)	
Occupational Status of Parent	Both Parents Work	43/(22.8)	68/(61.3)	0.000
	One of the Parents Works	133/(70.4)	42/(37.8)	
	Neither of the Parents Works	13/(6.8)	1/(0.9)	
Economic Security	Good	27/(6.9)	58/(52.3)	0.000
	Mediocre	69/(36.3)	45/(40.5)	
	Poor	94/(49.5)	8/(7.2)	
Type of Family	Complete	165/(86.8)	98/(88.3)	0.439
	Incomplete	25/(13.2)	13/(11.7)	
Religiosity	Practicing Catholic	153/(80.5)	100/(90.1)	0.108
	Non-Practising Catholic	32/(16.8)	10/(9.0)	
	Non-Believers	5/(2.7)	1/(0.9)	
Satisfaction with own Health Status	Very Satisfied	20/(10.5)	16/(14.4)	0.003
	Satisfied	80/(42.1)	47/(42.3)	
	Neither Satisfied nor Dissatisfied	66/(34.7)	43/(38.7)	
	Dissatisfied	24/(12.6)	2/(1.8)	
	Very Dissatisfied	0/(0.0%)	3/(2.7)	
Living Conditions	A House	130/(68.4)	89/(81.7)	0.012
	Block of Flats	60/(31.6)	20/(18.3)	
Separate room for a Child	Yes	101/(53.2)	78/(71.6)	0.001
	No	89/(46.8)	31/(28.4)	
The Number of People Taking Care of the Child	One	46/(24.2)	14/(12.8)	0.081
	Two	111/(58.4)	68/(62.4)	
	Three	26/(13.7)	20/(18.3)	
	Four	7/(3.7)	7/(6.4)	
The number of Children in the Family	One	37/(19.5)	31/(27.9)	0.086
	Two	67/(35.3)	44/(39.6)	
	Three	58/(30.5)	28/(25.2)	
	More than Three	28/(14.7)	8/(7.2)	
Feeling of Loneliness	Frequently	40/(21.1)		
	Sometimes	84/(44.2)		
	Never	66/(34.7)		

Values are presented as mean ± standard deviation; ^1^ M ± SD (R)*—medium ± standard deviation (range); ^2^ Mann–Whitney U test was used to compare the groups; ^3^ χ^2^ test was used to compare the groups.

**Table 2 ijerph-16-04173-t002:** Comparison of Hospital Anxiety and Depression Scale (HADS)-A and HADS-D between the study and control groups.

Group	HADS-A (Anxiety)				HADS-D (Depression)			
	Mean	Median	IQR	95% CI	Mean	Median	IQR	95% CI
Study	8.1	8	3.4	7.6–8.5	6.8	7	3.8	6,3–7,4
Control	4.7	4	3.1	4.1–5.3	3.7	3	3.3	3.1–4.3
*p* ^1^	0.0000				0.0000			

^1^*p*-value calculated using Mann-Whitney test.

**Table 3 ijerph-16-04173-t003:** Characteristics of the children in the study group.

Variable		M (SD) ^1^	(R) ^2^
Age of Children in Years		11.0 (4.3) 11.0 (4.3)	2–16
		N	%
Sex	Male	120	63.2
	Female	70	36.8
CP Type	Spastic Diplegia	98	51.6
	Spastic Hemiplegia	43	22.6
	Spastic Quadriplegia	26	13.7
	Other	23	12.1
GMFCS Level	I	25	13.2
	II	48	25.3
	III	35	18.4
	IV	44	23.2
	V	38	20.0
Diagnosed Degree of Intellectual Disability	Severe	16	8.4
	Significant	33	17.4
	Moderate	32	16.8
	Mild	29	15.3
	No Disability	80	42.1
Coexisting Symptoms ^3^	Speech Disorders	115	60.5
	Balance Disorders	109	57.4
	Intellectual Disability	110	57.9
	Epileptic Seizures	72	37.9
	Vision Disorders	55	28.9
	Behavioral Disorders	53	27.9
	Impaired Hearing	19	10.0
	No Symptoms	4	2.1

^1^ M (SD)—mean (standard deviation); ^2^ (R)—(Range); ^3^ the total does not add up to 100% because it was possible to choose any number of response options. CP: Cerebral Palsy; GMFCS: Gross Motor Function Classification System.

**Table 4 ijerph-16-04173-t004:** Regression model for predicting HADS-A.

Independent Variables	HADS-A (Anxiety)	
	*R*^2^ = 24.9% *F* = 32.9 *p* = 0.0000	
	*B*	*p*
Group (Study vs. Control)	2.82	0.0000
Health Status (Worse vs. Better)	1.24	0.0011
Financial Status (Bad vs. Good)	1.21	0.0066

**Table 5 ijerph-16-04173-t005:** Regression model for predicting HADS-D.

Independent Variables	HADS-D (Depression)	
	*R*^2^ = 28.0% *F* = 38.2 *p* = 0.0000	
	*B*	*p*
Group (Study vs. Control)	2.69	0.0000
Health Status (Worse vs. Better)	1.85	0.0000
Lack of Separate Room for a Child	2.03	0.0000

**Table 6 ijerph-16-04173-t006:** Regression model for predicting HADS-A only in the study group.

Independent Variables	HADS-A (Anxiety)	
	*R*^2^ = 11.4% *F* = 8.0 *p* = 0.0001	
	*B*	*p*
Perceived Support Available (pts)	−1.39	0.0004
Sex (Mother vs. Father)	1.11	0.0381
Financial Status (Bad vs. Good)	1.24	0.0100

**Table 7 ijerph-16-04173-t007:** Regression model for predicting HADS-D only in the study group.

Independent Variables	HADS-D (Depression)	
	*R*^2^ = 30.1% *F* = 19.9 *p* = 0.0000	
	*B*	*p*
Perceived Support Available (pts)	−1.99	0.0000
Currently Received Support (pts)	−1.05	0.0121
Financial Status (Bad vs. Good)	1.24	0.0091
Health Status (Worse vs. Better)	1.39	0.0044

**Table 8 ijerph-16-04173-t008:** Analysis of the relationship between factors linked to the parents of children with Cerebral Palsy (CP) and level of anxiety and depression in the study group.

Variables Related to Parent, Child and Environment		HADS-A Rang ^1^	*P* ^2^	HADS-D Rang ^1^	*P* ^2^
Age of Parent, in Years		−0.01	0.9189	0.17	0.0161
Sense of Coherence		−0.52	0.0000	−0.45	0.0000
Sense of Loneliness		0.43	0.0000	0.50	0.0000
Intellectual Disability of the Child		0.16	0.0250	0.01	0.8506
Social Support	Perceived Support Available	−0.27	0.0002	−0.48	0.0000
	Need for Support	−0.05	0.5096	−0.22	0.0020
	Support Seeking	−0.19	0.0104	−0.35	0.0000
	Currently Received Support	−0.14	0.0596	−0.37	0.0000
		HADS-A M (SD)	*P* ^3^	HADS-D M (SD)	*P* ^3^
GMFCS	I–III	7.8 (3.2)	0.1051	7.3 (3.8)	0.0916
	IV–V	8.4 (3.8)		6.3 (3.7)	
Attitude to Faith	Practicing Catholics	8.0 (3.4)	0.9065	6.6 (3.9)	0.0203
	Non-Practicing Catholics	8.1 (3.9)		8.1 (3.0)	
Education	Vocational	7.6 (3.5)	0.3325	7.8 (3.8)	0.0146
	Secondary	8.4 (3.7)		6.2 (3.6)	
	Higher	8.3 (2.8)		6.4 (3.8)	
The Number of People Taking Care of the Child	One	8.0 (3.3)	0.6722	7.9 (3.5)	0.0030
	Two	8.2 (3.7)		7.0 (3.7)	
	Three	7.7 (3.0)		5.1 (3.9)	
	Four	7.0 (2.6)		3.9 (3.4)	

^1^ Spearman’s correlation coefficient; ^2^
*p*- value for test probability calculated using the Spearman correlation coefficient M (SD)—mean (standard deviation); ^3^
*p*- value of test probability calculated with Kruskal-Wallis test.

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
