# Peer review of "Assessment of Anxiety and Depression in Polish Primary Parental Caregivers of Children with Cerebral Palsy Compared to a Control Group, as well as Identification of Selected Predictors"

_ijerph, 2019, doi:10.3390/ijerph16214173_

Round 1
Reviewer 1 Report
The authors assess the intensity of anxiety and depression symptoms in caregivers of children with cerebral palsy compared to a control group as well as identification of selected mental health predictors. With selected mental health predictors, the authors revealed that the parents of children with cerebral palsy presented visibly higher level of anxiety (explained variance, R squared 11.4-24.9%) and depression (R squared 28.0%-30.1%).
The study is performed properly and the appropriate statistical analysis were used.
Minor Comments:
It should be corrected.
Table 7, it should change HADS-A (depression) to HADS-D (depression).
Table 8, it should change HAD-A to HADS-A and HAD-D to HADS-D.
Is it typing error? Comma as decimal point presented in Tables 2, 4, 5, 6, 7, 8.
Author Response
Dear Reviewer,
thank you very much for the helpful comments, we have included the corrections in text. In the attached file we briefly respond to the suggestions.
Yours sincerely Barbara Gugała
Response to Reviewer 1 Comments
Thank you very much for the helpful comments and suggested changes which will improve the quality of the article. We have carefully reviewed the manuscript taking into account all the comments. Below, we briefly respond to suggestions.
Additionally, we amended the authors’ affiliations, to account for a change in the organisation of the University of Rzeszów implemented recently.
Point 1: Table 7, it should change HADS-A (depression) to HADS-D (depression).
Response 1: In table 7, HADS-A was corrected for HADS-D
Point 2: Table 8, it should change HAD-A to HADS-A and HAD-D to HADS-D.
Response 2: In table 8, HAD-A to HADS-A and HAD-D to HADS-D were corrected.
Point 3: Is it typing error? Comma as decimal point presented in Tables 2, 4, 5, 6, 7, 8.
Response 2: In tables 2, 4, 5, 6, 7, 8 the decimal points have been corrected as dot
Reviewer 2 Report
The aim of this study was to assess the intensity of anxiety and depression 14 symptoms in caregivers of children with cerebral palsy (CP) compared to a control group as well as 15 identification of selected mental health predictors. The control group seemed to be the parents of children without CP. However, this is not clearly reflected in the title and the manuscript. The authors could revise the title to reflect this. In saying that, it raises a concern around comparability – are the caregivers blood-related as parents are? If the authors used the word caregivers and parents interchangeably, it is suggested that a consistent definition be used throughout. If not, the authors may wish to justify your choice of such comparisons.
Other suggestions are:
Heading 2.2 “Qualification” referred to “Eligibility” to participate in the study. Please revise. Section 2.4 Clinical parameters does not seem to be relevant to the main aim, i.e. mental health of the parents. This can be omitted. Could the authors add a sentence or two regarding the validating and reliability of the specially designed questionnaire (Section 2.5.2) please? Section 2.6 Statistical analyses: The authors mentioned that means and SD were reported but opt to use Kruskal-Wallis test, which is a non-parametric test for skewed variables where median and interquartile range should be used to describe the variables. Please revise. On a separate note, it is unclear if the assumptions for multiple linear regression (MLR) were met. Perhaps if there are clinically validated cut-off points for the mental health outcome measures, the authors could use logistic regression models instead if the assumptions for MLR were not met. The authors could also report the relative risk of poor mental health outcomes. Table 1: Please revise p-values of .0000 to <0.001 The discussion and conclusion are well-posited around the current findings albeit revisions may be required following changes of the statistical analyses as recommended above, where necessary. The overall message aligned with a publication around unmet needs (financial and social in a broader context) among family members and the mental toll on family members, of which the authors may wish to cite (DOI: 10.1186/s12888-017-1483-z). The authors could also note the limitations of such cross-sectional survey.Author Response
Dear Reviewer,
Thank you very much for the helpful comments and suggested changes which will improve the quality of the article. We have carefully reviewed the manuscript taking into account all the comments. In the attached file we briefly respond tothe suggestions.
Yours sincerely Barbara Gugała
Dear Reviewer,
Thank you very much for the helpful comments and suggested changes which will improve the quality of the article. We have carefully reviewed the manuscript taking into account all the comments. Below, we briefly respond to suggestions.
Additionally, we amended the authors’ affiliations, to account for a change in the organisation of the University of Rzeszów implemented recently.
Point 1: English language and style: Moderate English changes required
Response 1: The article has been revised by a native-speaking translator who has corrected the phrasing in English. The changes are marked in red.
Point 2: The control group seemed to be the parents of children without CP. However, this is not clearly reflected in the title and the manuscript. The authors could revise the title to reflect this. In saying that, it raises a concern around comparability – are the caregivers blood-related as parents are? If the authors used the word caregivers and parents interchangeably, it is suggested that a consistent definition be used throughout. If not, the authors may wish to justify your choice of such comparisons.
Response 2: Since all the subjects in the study group were biological parents of the children, we have replaced the term ‘caregiver’ with the term ‘parent’ (or another word matching the context) in the manuscript, in line with the Reviewer’s suggestion. However, the study focuses on those parents who spend more time than their spouses taking care of their children with CP (Line 102-103), hence in the title and in descriptions of the study group we have decided to use the phrase ‘primary parental caregivers’. In the manuscript the changes are marked in red.
Point 3: Heading 2.2 “Qualification” referred to “Eligibility” to participate in the study. Please revise.
Response 3: The change has been made (‘Eligibility to participate in the study’). The text is marked in red.
Point 4: Section 2.4 Clinical parameters does not seem to be relevant to the main aim, i.e. mental health of the parents. This can be omitted.
Response 4: The method of acquiring the clinical parameters (Section 2.4) is important for the characteristics of the children (Table 3), and the clinical parameters were used as children-related variables possibly differentiating the levels of anxiety and depression observed in the parents , Table 8 (Analysis of the relationship between factors linked to the parents of children with CP and level of anxiety and depression in the study group). Therefore we would like to leave Section 2.4 as it is.
Point 5: Section 2.6 Statistical analyses: The authors mentioned that means and SD were reported but opt to use Kruskal-Wallis test, which is a non-parametric test for skewed variables where median and interquartile range should be used to describe the variables. Please revise.
Response 5: As suggested by the Reviewer, we replaced means and SD with median and interquartile range statistics. Tables 1 and 2 have been changed to include median and IQR. The description of statistical methods (Section 2.6.) has been accordingly (as highlighted in red).
Point 6: On a separate note, it is unclear if the assumptions for multiple linear regression (MLR) were met. Perhaps if there are clinically validated cut-off points for the mental health outcome measures, the authors could use logistic regression models instead if the assumptions for MLR were not met.
Response 6:
Thank you for this suggestion related to the use of logistic regression models, but we checked the assumptions for regression models (mainly the distribution of residuals, with regard to possible outliers which possibly would affect the values of parameters in the identified models). Distribution of residuals was highly regular, approaching normality. To illustrate this, below we are enclosing two graphs showing the residuals for the model presented in Table 4. Because they are not relevant to medical practice, these results were not included in the article.
Point 7: Table 1: Please revise p-values of .0000 to <0.001
Response 7: In Table 1, p-values of .0000 were corrected to <0.001
Point 8: The overall message aligned with a publication around unmet needs (financial and social in a broader context) among family members and the mental toll on family members, of which the authors may wish to cite (DOI: 10.1186/s12888-017-1483-z).
Response 8: A reference to the impact of financial situation on family health is provided in Line 403, supported by Line 587 in source material.
Point 9: The authors could also note the limitations of such cross-sectional survey.
Response 9: Limitations of the current study have been added (Line 436)